# Segregated Conductive Polymer Composite with Fe_3_O_4_-Decorated Graphite Nanoparticles for Microwave Shielding

**DOI:** 10.3390/ma17122808

**Published:** 2024-06-08

**Authors:** Ludmila Yu. Matzui, Oleksii A. Syvolozhskyi, Ludmila L. Vovchenko, Olena S. Yakovenko, Tetyana A. Len, Olena V. Ischenko, Anna V. Vakaliuk, Victor V. Oliynyk, Volodymyr V. Zagorodnii, Antonina Naumenko, Maria Cojocari, Georgy Fedorov, Polina Kuzhir

**Affiliations:** 1Facultiy of Physics, Taras Shevchenko National University of Kyiv, Volodymyrska Str. 64/13, 01601 Kyiv, Ukraine; mail.olexiy@gmail.com (O.A.S.); vovch@univ.kiev.ua (L.L.V.); alenka-ya@ukr.net (O.S.Y.); intercalant@univ.kiev.ua (T.A.L.); ovv1@univ.kiev.ua (V.V.O.); a_naumenko@univ.kiev.ua (A.N.); 2Facultiy of Chemistry, Taras Shevchenko National University of Kyiv, Volodymyrska Str. 64/13, 01601 Kyiv, Ukraine; elischenko58@gmail.com (O.V.I.); anna.vakaliuk@gmail.com (A.V.V.); 3Department of Physics and Mathematics, University of Eastern Finland, Yliopistokatu 7, FI-80101 Joensuu, Finland; mcojocar@uef.fi (M.C.); georgy.fedorov@uef.fi (G.F.); polina.kuzhir@uef.fi (P.K.)

**Keywords:** composites with hybrid filler, a segregated structure of conductive filler, electromagnetic interference shielding materials

## Abstract

Graphite nanoplatelets (GNPs)—the segregated ultra-high molecular weight polyethylene (UHMWPE)-based composites with hybrid filler—decorated with Fe_3_O_4_ were developed. Using X-ray diffraction and scanning electron microscopy, it was shown that the decorated component has the shape of separate granules, or their clusters were distributed evenly over the GNPs surface. The individual Fe_3_O_4_ nanoparticles are predominantly rounded, with diameters of approximately 20–60 nm. The use of GNPs/Fe_3_O_4_ as a filler leads to significant decreases in the percolation limit φc, 0.97 vol% vs. 0.56 vol% for GNPs/UHMWPE- and (GNPs/Fe_3_O_4_)/UHMWPE segregated composite material (SCM), respectively. Modification of the GNP surface with Fe_3_O_4_ leads to an essential improvement in the electromagnetic interference shielding due to enhanced microwave absorption in the 26–37 GHz frequency range in its turn by abundant surface functional groups and lattice defects of GNPs/Fe_3_O_4_ nanoparticles.

## 1. Introduction

The rapid development of electronic technology and communications and the expansion of the frequency range covered lead to a high level of electromagnetic pollution [1,2], which might have an effect on the human body and the environment [3,4]. Thus, the development of highly efficient electromagnetic interference (EMI) shielding materials is an important challenge. In that respect, conductive polymer composites (CPCs) have many advantages over traditional metal-based materials for EMI shielding applications, including being lightweight, having tunable electrical conductivity, having corrosion resistance, and having facile processability [5,6,7]. Using nanocarbon, in particular carbon nanotubes (CNTs), graphene, and graphite nanoparticles, as fillers is very attractive because of their high aspect ratios, which refer materials with high conductivity under a percolation threshold [7,8,9,10,11]. However, the absorption of electromagnetic radiation in such composites is ensured exclusively due to the dielectric losses. It means that the high shielding effectiveness (SE) of such conductive materials is determined by the reflection mechanism, which might be a subject of potential secondary pollution [10,11,12,13]. Normally, it is difficult to reach high shielding efficiency in composites containing a single conductive or magnetic inclusion. The combination of two or more fillers providing dielectric and magnetic losses has the benefit of efficiently tuning electromagnetic parameters and allowing the fabrication of materials demonstrating synergistic absorption effects. 

The use of hybrid magnetodielectric fillers contributing to magnetodielectric losses is a popular strategy for obtaining highly absorptive materials [14,15,16,17].

The decoration of graphene nanoplatelets and graphitic nanoparticles with a large surface area with magnetic metals is the most flexible way to enhance microwave absorption. As shown in a number of works, the use of conductive nanocarbon with magnetic components, such as FeCo [18], Ni_0.6_Zn_0.4_Fe_2_O_4_ [19], barium hexaferrites [20,21,22], Fe_2_O_4_ [23], Fe [24], Fe_3_O_4_ [25,26], and Co_3_O_4_ [27], significantly increases absorption, due to their magnetic losses. For example, Zong et al. [26] reported that the RGO–Fe_3_O_4_-3 composites with the maximum reflection losses (RL) at 6.6 GHz reached 44.6 dB for the 3.9 mm thick protective layer, while with a 2.0 mm thickness, the RL bandwidth with less than 10 dB can reach up to 4.3 GHz (from 12.2 to 16.5 GHz). Using Co3O4-functionalized MoS_2_ nanosheets, Chai et al. managed to achieve a shielding effectiveness of −41.2 dB at 12 GHz for 10 wt% filler content [27]. P. Manjappa et al. [23] synthesized composites of α-Fe_2_O_3_/MWCNT/graphene in an LDPE polymer matrix with superior dielectric, magnetic, and ohmic losses, allowing a 40 dB (99.99%) attenuation for 5% α-Fe_2_O_3_ in the X-band. 

Another possibility for enhancing shielding efficiency is manipulating the structural design [28,29,30]. Recently, the method of segregated structure of the conductive filler in the polymer matrix construction was widely acknowledged for achieving excellent conductive performance and shielding efficiency with a lower functional filler loading [31,32,33,34,35]. In these structures, the percolation threshold was reduced significantly by the selective location of the conductive fillers at the polymer granule interfaces and by the formation and connection of the conductive pathways to construct dense networks. For example, the electrical conductivity of the segregated GNPs–polypropylene composites improved by 20.79 S/m at a low filler content (1 wt%) [36]. A comparative study of the segregated and random CNT/polyethylene (PE) composites revealed that the electrical conductivity of 3 wt% CNTs loaded with the segregated composite was more than two orders higher than that of composites with random distribution fillers [37]. The SCMs for EMI shielding applications can be prepared by hot pressing polymer particles coated with conductive fillers on the surface at high temperatures. These structural designs effectively solve the problem of the high percolation threshold of CPCs and manage other important performances of EMI shielding materials.

These SCMs tend to support multiple reflections and scatterings to extend the transmission path of the EM wave, leading to the dissipation of more EM energy. 

Currently, the conductive fillers used for segregated EMI shielding composites can be divided into several types: metallic nano/micro-fillers [38], micro-scale conductive fillers (graphite and carbon black) and nano-scale conductive fillers (graphene, MWCNTs, and MXene) [39,40,41,42], and hybrid fillers (graphene + MWCNTs and nanocarbon decorated with magnetic nanoparticles) [43,44,45,46].

In this study, segregated composites were produced using graphite nanoparticles decorated with Fe_3_O_4_ nanoparticles, and their electrical and shielding properties were studied at different filler contents in the composite. The choice of iron oxide among various types of magnetic materials as a magnetic component was determined by its stable and high magnetic characteristics and a fairly simple procedure of chemical modification of GNPs.

## 2. Materials and Methods

### 2.1. Materials

Ultra-high molecular weight polyethylene (UHMWPE) has been widely studied and utilized in microwave shielding applications, as it is chemical resistant, thermally stable, and easily processed. Additionally, PE-based materials have been investigated due to their possible applications in 5G technology, where reliable and efficient microwave shielding is crucial for preventing interference and ensuring network performance. In this work, UHMWPE (H(CH_2_CH_2_)nH was used as the polymer matrix of segregated systems and had a density of 0.94 g/cm^3^; UHMWPE particles have a globular shape and an overall particle size of 100 μm. In order to prepare graphite nanoplatelets (GNPs), oxidized natural graphite was thermo-exfoliated at 1000 °C. After ultrasonic treatment of thermo-exfoliated graphite particles in acetone for 3 h using a Baku 9050 (Baku, China) apparatus with a maximum output power of 50 W and an ultrasound intensity of 40 kHz [47], GNP particles of 15–45 nm (thickness distribution reached 30 nm) were obtained. The lateral dimensions of the GNPs particles were 1–10 µm [47].

### 2.2. Preparation Methods for GNPs Decorated by Fe_3_O_4_

An amount of 2.003 g of the GNPs powder was filled with iron nitrate solution (0.400 g of Fe dissolved in ~20 mL of HNO_3_). This was followed by 1 h of stirring the mixture and its evaporation to dryness at 120 °C. The final step was thermal decomposition in an argon atmosphere for 1 h at a temperature of 600 °C. The scheme of preparation of the GNPs decorated by Fe_3_O_4_ are presented in Figure 1.

### 2.3. CMs Preparation

Segregated UHMWPE-based composites with graphite nanoparticle-decorated Fe_3_O_4_ nanoparticles were prepared by the hot compacting of particles covered by the GNPs layer [35]. The GNPs layer’s thickness had a proportional concentration in the composite. In order to obtain UHMWPE globules covered by GNPs, the polymer and nanocarbon powders were mixed in the IKA ULTRA TURRAX Tube Drive homogenizer test tube with a stirrer, with 3000 rpm as the rotation speed. Then, the mixture was hot-compacted for 5 min at 50 MPa in an enclosed hot die heated to 160 °C, after which it was cooled down to room temperature. The GNPs/UHMWPE and (GNPs/Fe_3_O_4_)/UHMWPE composites with 0.5–5 vol% of GNPs or the (GNPs Fe_3_O_4_) with the segregated distribution of the filler were obtained using this procedure.

### 2.4. Experimental Methods

Optical microscopy was used to study the filler particles and epoxy composite structure and morphology (“Mikmed-1” with ETREK PCM-510 attachment). 

The materials surfaces were studied with a Tescan scanning electron microscope. The elemental analysis of the samples was performed using energy-dispersive spectroscopy (EDS); X-ray diffraction (XRD) analysis was used to examine the phase composition of the prepared GNPs decorated by Fe_3_O_4_. XRD investigations were performed using a DRON-4–07 X-ray diffractometer with Co Kα-filtered radiation (λ = 1.7902 Å) at room temperature. 

Raman spectra were obtained using Renishaw in Via with 2400 lines/mm grating and a ×50 objective at an excitation wavelength of 514 nm. The chemical composition of fillers was determined by Fourier-transform infrared spectrometry (FTIR) using the IRTracer-100 FTIR spectrometer (Shimadzu Scientific Instruments, Japan)). The standard two-probe method in DC mode at room temperature with a measurement limit of electrical resistance of 10^10^ Ω was used for the electrical conductivity measurement of the investigated composites. Resistances above 10^10^ Ω were measured by a Е6–13 teraommeter.

The electromagnetic shielding characteristics of the composites were examined in the frequency band of 26–37 GHz at room temperature utilizing the P2-65 scalar microwave network analyzer. The rectangular waveguides exhibited the subsequent cross-sectional dimensions: 7.2 × 3.4 mm^2^. Specimens were obtained from the blocks of isolated GNPs/UHMWPE and (GNPs/Fe_3_O_4_)/UHMWPE composites by employing a milling apparatus with a numerical control positioning resolution of 0.025 mm. The configuration of the analyzed specimens ensured the full occupation of cross-sectional areas of the waveguides, whereas their thickness measured 1 mm.

The total shielding efficiency (*SE_T_*), reflection (*SE_R_*), and absorption (*SE_A_*), as well the absorption (*A*), transmission (*T*), and reflection (*R*) coefficients, for an incident EM wave on a layer of EMI shielding material were calculated using microwave scattering parameters (S11 and S21) using Equations (1)–(6) [48]:(1)$$R=S_{11}^{2},$$
(2)$${SE}_{R}=10log(1-R),$$
(3)$$T=S_{21}^{2},$$
(4)$${SE}_{T}=10log(T),$$
(5)A=1−R−T,
(6)$${SE}_{A}=10log\frac{T}{1-R},$$
where EMR transmission (T) and reflection (R) indices were determined as $R={\left|E_{R}/E_{I}\right|}^{2}$ and $T={\left|E_{T}/E_{I}\right|}^{2}$ and where $E_{I}$, $E_{R}$, and $E_{T}$ are the electric field strengths of the incident, reflected, and transmitted waves, respectively. 

## 3. Results

### 3.1. Material Characterization

As-received GNPs decorated by Fe_3_O_4_ were characterized using SEM and X-ray instruments. 

The results of the SEM study (at different magnifications) on the morphology of NPs decorated by Fe_3_O_4_ powder are presented in Figure 2. As seen in Figure 2, the metal is in the form of separate granules or their clusters and is evenly distributed over the surface of the GNP plates. The individual Fe_3_O_4_ nanoparticles are predominantly rounded, with diameters varying from 20 to 60 nm (Figure 2a). Most of the nanoparticles agglomerate in clusters of different shapes (Figure 2b). The sizes of the nanoparticle agglomerations range from 200–1000 nm. As one can see from Figure 2e,f, the distribution of Fe_3_O_4_ nanoparticles in the GNP nanoparticles volume is rather uniform. Individual GNP Fe_3_O_4_ nanoparticles (or their agglomerations) are located not only on the plane surfaces of GNP particles but also on its edges and the pores between GNP particles (Figure 2c,d).

The phase composition of the synthesized composites studied by X-ray diffraction and the XRD pattern of the as-received Fe_3_O_4_ powder are presented in Figure 3a. The diffraction peaks at 33.52, 57.01, and 70.46, correspond to the graphitic structures (002), (101), and (004). Decoration results in several new peaks appearing, which could be indexed as the characteristic (110), (220), (311), (222), (400), (422), (511), and (440) reflections of the cubic spinel crystal structure of Fe_3_O_4_. Additionally, the Fe_3_O_4_ phase has low, intense reflections, corresponding to the Fe_2_O_3_ phase also presented.

Fourier-transform infrared (FTIR) spectroscopy is an important and powerful method for material characterization that can be used to identify the functional groups present in the polymer composites. By comparing the FTIR spectra of the composites with the reference spectra of known compounds, the presence of functional groups from both the polymer matrix and the GNPs decorated by Fe_3_O_4_ can be identified. This helps to confirm the successful covering of the graphite nanoparticles and the incorporation of these nanoparticles into the polymer matrix.

The FTIR spectra of pure GNPs, Fe_3_O_4_, polyethylene (PE), and SCMs with (GNPs/Fe_3_O_4_)/UHMPE are shown in Figure 3b. The obtained FTIR spectra of the composite show a rich set of vibrational bands, which, as the analysis showed, are both characteristic of the moieties of the composite and inherent in the composite itself. From these spectra (lines 1, 3, and 5), one can identify three intensive, well-pronounced doublets around 750, 1470, and 2900 cm^−1^, which correspond to the C-H rocking, bending, and stretching vibrations of the polyethylene matrix, respectively; the weak band at ~1300 cm^−1^ is the twisting deformation band of PE [49]. The appearance of a strong band at 635 cm^−1^ shows the formation of Fe-oxide nanoparticles. This band is complicated and can be fitted on two bands with maxima at 569 cm^−1^ and 635 cm^−1^, which reflect stretching and bending vibrations of the Fe-O bond of Fe_3_O_4_ nanoparticles (lines 3–5) [50]. The band at 480 cm^−1^ may point to the presence of vibrations of Fe-O, and the peak at around 850–900 cm^−1^ corresponds to the O-Fe-O stretching vibration [51]. The peaks at 1369 cm^−1^ and 1050 cm^−1^ are assigned to the C-OH stretching vibration and C-O vibrations [52]. The characteristic graphite peak at 1630 cm^−1^, assigned as the C=C vibration in graphene nanoparticles, has also been detected. In the experimental spectra, the -OH vibration of intercalated water in the region 3350–3680 cm^−1^ [53] and the very weak band corresponding to the C=O stretching vibration (1732 cm^−1^) [54] were also detected. 

Our results show the shift in peak position, the broadening of the bands, and changes in the intensity of bands related to functional groups in pure materials and those in the hybrid nanocomposite. This can suggest interactions between the nanoparticles and the polymer and can provide insights into the compatibility and dispersion of nanoparticles within the polymer matrix.

As it is known, Raman spectroscopy is a powerful and nondestructive method used to characterize carbon-based materials, such as graphene, carbon nanotubes, graphene-like structures, and graphites with different structure perfection degrees [55]. The spectrum of monocrystalline ideal graphite contains only one intensive band, the so-called G-band (ν^−1^ = 1580 cm^−1^ and half-width Δν = 12.5 ± 0.5cm^−1^) [56]. Less-ordered graphites exhibit an additional line in the region of ~1350 cm^−1^ in their Raman spectra. This is the D-band appearing due to defects, the finite size of crystallites, disorder, and the re-hybridization of carbon atoms from the sp^2^ state to the sp^3^ state. The higher the defectiveness of the graphite structure, the more intense its D-band. In [57], it was clear that D peaks consist of two components at 1350 cm^−1^ and 1375 cm^−1^. The redistribution of Raman intensity between these components also depends on the degree of structural disorder. Therefore, the position of the G-band line is shifted from 1570 cm^−1^ to 1585 cm^−1^ [56]. The intensity ratio of the G-band to the D-band (*I_G_/I_D_*) is usually a measure of sp^2^/sp^3^ [58]. In addition, the Raman spectra of carbon materials are very sensitive to changes in the environment, for example, when they are embedded in a polymer matrix [59].

The results of the study on Raman spectroscopy for GNPs and GNPs decorated by Fe_3_O_4_ are presented in Figure 3c,d. As shown in Figure 3c,d, two signal peaks emerged at ~1350 cm^−1^ (D-band) and ~1580 cm^−1^ (G-band) for both types of specimens. The heterogeneity of the structure of GNPs, according to the Raman spectrum, is rather pronounced. As one can see from Figure 3. the Raman spectra obtained for different points of the sample are very different. In certain parts of the sample, the intensity of the Raman spectrum is close to the Raman spectrum of pure graphite. In any part, the ratio of *I_G_/I_D_* is as high as 11.11; for other parts of the investigated samples, the ratio of *I_G_/I_D_* is 1.64. Such a high intensity of the D-line can appear due to the activation of the first-order scattering process of the sp^3^ carbon that indicates an increase in the number of defects in graphite layers or the reduction of the size of the crystallites along the graphite planes.

The intensity ratios of the G-band to the D-band (*I_G_/I_D_*) for GNPs decorated by Fe_3_O_4_ (Table 1) range from 2.38–3.2 for different spots on the samples, suggesting that there is no essential reduction in the average size of the sp^2^ domains observed under the chemical modification of GNPs by Fe_3_O_4_

### 3.2. Electrical Conductivity

The electrical conductivities of SCMs with GNPs/UHMWPE and SMCs with (GNPs/Fe_3_O_4_)/UHMWPE composites with different filler volume fractions are presented in Figure 4.

As one can see from Figure 4, the sharp increase in the electrical conductivity occurred in the SCMs with (GNPs/Fe_3_O_4_)/UHMPE, with the filler loading increasing from 0.4 to 5 vol%, showing a distinct percolation behavior. It is necessary to admit a sharper increase in the conductivity of SCMs with GNPs/Fe_3_O_4_/UHMWPE, compared to SCMs with pure GNPs, with an increase in the filler content in the composite. 

The behavior of electrical conductivity (σ) in the region of the insulator–metal transition can be described within the framework of the classical theory of percolation, and the corresponding expression has the form [60]:(7)$$\sigma =\sigma_{0}{\left(\varphi -\varphi_{c}\right)}^{t}\;at\;\varphi >\varphi_{c};$$
where *ϕ_c_* is the percolation threshold, *ϕ* is the volume fraction of the fillers, and *t* is the critical exponent. The percolation threshold was obtained from the experimental dates. As seen from the insertion in Figure 4, dependences log σ vs log (φ − $\varphi_{c})$) are linear, obtaining the parameters *t* and σ_0_. Percolation parameters *ϕ_c_*, *t*, and $\sigma_{0}$, for investigated composites are given in Table 2. As one can see, the use of (GNPs/Fe_3_O_4_)/UHMWPE in SCMs as a filler essentially decreases the percolation threshold *φ_c_*. The calculated percolation thresholds are 0.97 vol% and 0.56 vol% for the SCMs with GNPs/UHMWPE and SCMs with (GNPs/Fe_3_O_4_)/UHMWPE samples, respectively. Therefore, the chemical modification of GNPs by Fe_3_O_4_not only reduces the percolation threshold significantly but also increases the value of electrical conductivity (see Figure 4). It is known that in the process of chemical modification, a significant delamination of GNP particles takes place [61,62], and as a result, there is a significant increase in the aspect ratio of GNPs. This leads to a decrease in the percolation limit and a decrease in electrical conductivity at the same concentration of fillers in the composite [63,64].

In addition, the analysis of the changes in the critical index t in SCMs with GNPs/UHMWPE (2.78), compared to the SCMs with (GNPs/Fe_3_O_4_)/UHMWPE samples (1.63), suggest that the three-dimensional conductive network is not essentially changed under the modification of the GNP particles by Fe_3_O_4_ [65].

### 3.3. Microwave Properties

For a fully comprehensive analysis of the electrodynamic characteristics of composites with a segregated structure of the conducting network, namely the coefficients of transmission, reflection, and absorption of EMI in the range of 26–37 GHz, the composite SCMs with (GNPs-Fe_3_O_4_)/UHMWPE were experimentally investigated. The results are shown in Figure 5. It can be seen that SE_T_ and SE_A_ weakly depend on the frequency in the studied frequency range of 25–37 GHz. The feature appearing in the range of 33–34 GHz on the EMI SE curves for all investigated samples, regardless of the filler content or type, occurs due to the special anisotropic segregated structure [66]. It was shown that multiple reflections of microwaves in samples with a segregated structure of a conducting cluster at the boundaries of the polymer cell-conducting layer are additional absorption mechanisms in segregated arrays. Increased pressure during the formation of CM with a segregated structure leads to the structural anisotropy of the polymer cell and multiple reflections of electromagnetic waves of a certain frequency in an oriented cell, which is the reason for the appearance of a resonant peak due to the design intervention. In our case, SCMs were obtained at a pressure in the cell of 50 MPa, which is higher than the commonly used pressing pressures (10 MPa), so it is quite logical to assume that the observed anomalies in the frequency dependences are associated with the constructive interference induced by the structural anisotropy of the polymer cell in the formation process. 

We therefore consider the averaged SE (over the frequency) values and analyze their dependence on the filler concentration and type.

The shielding efficiency of the EMI (SE) of a material is determine by the reflection and absorption. An impedance mismatch between the air (vacuum) and the material results in the reflection of EMI. The absorption EMI is mainly connected with ohmic losses, polarization losses, and magnetic losses coming from magnetic particles in the EM field. Figure 6a presents the loading dependences of microwave transmission (SE_T_), absorption (SE_A_), and reflection (SE_R_) for segregated SCMs with GNPs/UHMPE and segregated SCMs with (GNPs-Fe_3_O_4_)/UHMPE at a frequency of 30 GHz. We note a significant increase in SE_T_ in SCMs with (GNPs/Fe_3_O_4_)/UHMPE in comparison with SCMs with GNPs/UHMPE. Notably, SPCs with the GNPs/Fe_3_O_4_ filler demonstrate a much higher SE_T_ value than composites with pure GNP conductive fillers. As one can see from Figure 6a, the transition coefficient decreases more sharply with the increase in the loader content for SCMs with GNPs/Fe_3_O_4_/UHMWPE than for SCMs with GNPs/UHMWPE. Even at 5 vol% filler in SCMs with (GNPs/Fe_3_O_4_)/UHMWPE, T is 1.3 × 10^−5^, which is more than three orders of magnitude lower than for SCMs with GNPs/UHMWPE with the same filler concentration (3.5 × 10^−2^).

Another important trend seen from Figure 6a is increasing SE_T_ and SE_A_ with filler loading, whereas SE_R_ remains low when filler loading increases to up to 3 vol% of fillers and becomes practically independent of the type of SCMs. Therefore, the value (SER)3%/(SER)0.5% is equal to 1.1, while (SEA)3%/(SEA)0.5% = 80. Furthermore, the SEA values are much higher than the SER values, and the difference between SEA and SER values increase with increased fillers. This indicates that radiation absorption determines the electromagnetic shielding for investigated composites. 

The *absorption coefficient* A, defined as the ratio of the *absorbed energy* to the incident power of the EMI, does not reflect the ability of the sample to absorb EMI. The effective material absorption Aeff in the total shielding efficiency can be determine as
(8)$$A_{eff}=\frac{1-R-T}{1-R}$$
and the results are presented in Figure 6b. 

The dates presented in Figure 6b show that the EMR absorption index increases *A_eff_* with the filler content for both types of SCMs. The change in the absorption index with increasing concentration correlates with the type of dependence of the electrical conductivity curve of the studied SCMs. The results presented in Figure 6b also demonstrate that for SCMs with (GNPs/Fe_3_O_4_)/UHMPE under a filler content ≥2 vol%, the microwave absorption reaches 0.95–0.99, while for (GNPs)/UHMPE, *A_eff_* reaches 0.98 at a much higher GNP concentration—5 vol%. Thus, the magnetic component in composites with GNPs/Fe_3_O_4_ essentially increases the absorption part of shielding.

It is known that shielding by reflection (*SE_R_*) for relatively thick (thicker than skin depth) conducting materials is [67]:(9)$$SE_{R}=39.5-20\log\frac{\sigma }{2\pi f\mu }$$

Absorption is one more mechanism that influences the overall magnitude of EMI shielding efficiency, and A depends on the shield thickness d. The presence of electric and/or magnetic dipoles in the material that interact with electromagnetic waves enhances shielding efficiency by absorption (*SE_A_*):(10)$$SE_{A}=20\log e^{d/\delta },$$
where the skin depth *δ* is
(11)$$\delta ={\left(\sqrt{\pi f\mu \sigma }\right)}^{-1},$$
*f* is the frequency, *σ* is the electrical conductivity, *μ_r_* is the magnetic permeability, and *d* is the thickness of the shield.

Another important shielding mechanism is related to the multiple reflection of electromagnetic waves within the material: (12)$$SE_{MR}=20\log|1-e^{\frac{d}{\delta }}|.$$

Using our experimental values of SE_R_ and SE_A_ for SCMs with (GNPs/Fe_3_O_4_)/UHMPE, we can calculate the loading dependences of *σ_ac_* and *μ_r_*. The results are presented in Figure 6c.

As can be seen from Figure 6c, the change in SE_R_, with an increase in the filler concentration in the composite, is determined by the change in the value of *σ_ac_/μ_r_,* and the weak dependence of SE_R_ for SCMs with (GNPs/Fe_3_O_4_)/UHMPE is associated with simultaneous increases in the concentrations of both *σ_ac_* and *μ_r_*. The absorption loss is proportional to the value *of σ_ac*_μ_r_*, indicating that the higher conductivity, as well as higher magnetic permeability, leads to a higher absorption loss.

## 4. Discussion

It is known that for CMs containing both the conductive and magnetic components, adsorption originates from dielectric and magnetic tangent losses and natural resonance. The results presented in the previous section unambiguously indicate that the presence of the Fe_3_O_4_ nanoparticles enhances the CM absorptivity of the electromagnetic radiation in the 20 to 36 GHz range. The dielectric losses originate from the dipole polarizations and interfacial polarizations. In the composite sample with the conduction electrons and charged particles, some energy is dissipated as heat in the interaction with the incident electromagnetic radiation. In this case, an electric part of the incident electromagnetic wave interacts with polarization centers, which are formed on the defects in GNPs and Fe_3_O_4_, nanoparticles, and abundant surface functional groups. When a three-dimensional conducting network is formed in segregated CMs, numerous charged and conducting interfaces are formed at the boundary of the cellular structure, which are capable of absorbing and reflected EMI.

In the investigated composite, we used GNPs decorated by Fe_3_O_4_ nanoparticles as the conductive filler, and under the formation of the segregated structure from UHMWPE granules coated by this filler, a lot of interphase areas between Fe_3_O_4_ and GNPs nanoparticles and UHMPE globules appeared, which were the sources of multiple reflections of EMR inside the sample that respectively increased the absorption of EMR and the effective shielding of such a structure. The increase in filler concentration in CMs results in the increase in conductivity, due to the formation of an additional number of conductive networks and the increase in their own conductivity. It is known that dielectric loss is determined by the imaginary permittivity of CMs, and it is primely connected with conductivity (*ε*″ = *σ*⁄2π*fε*_0_). Therefore, the increase in conductivity under increasing filler content leads to the increase in imaginary permittivity materials (see Figure 6c) and, respectively, also increases the absorption of EMR. 

The magnetic losses in doped substrates are mainly determined by magnetic hysteresis, magnetic resonance domain wall displacement, and eddy current effects. Keeping in mind that that the magnetic field of the incident EM waves is very weak, we can conclude that the contribution of hysteresis loss is insignificant; the domain wall resonance usually occurs at a much lower frequency (megahertz). That is why it is supposed that natural ferromagnetic resonance and eddy current effects are two basic loss mechanisms in the gigahertz range for ferromagnetic absorbers. In developed CMs, the magnetic particles, due to their low content in the composite, may not greatly contribute to the magnetic loss/complex. However, it is expected that, due to abundant surface functional groups and lattice defects on GNPs/Fe_3_O_4_ nanoparticles and to the synergistic effect between Fe_3_O_4_ nanoparticles and GNPs, there is an enhancement of microwave absorption performance as a result of essentially increasing the inter-facial scatterings, and corresponding complex permittivity appears.

## 5. Conclusions

To conclude, we succeeded in fabricating composite materials in which segregated graphite nanoplatelets decorated with Fe_3_O_4_ nanoparticles were evenly dispersed in the polyethylene matrix. The structure and morphology of the novel materials had electrical and magnetic properties that ensured very efficient electromagnetic shielding of microwave radiation. Analyses of both DC and AC electromagnetic responses showed that the decoration of the GNPs with Fe_3_O_4_ nanoparticles led to a significant decrease in the percolation limit *φ_c_* within the entire studied frequency range. They were found to be 0.97 vol% and 0.56 vol% for samples of SCMs with GNPs/UHMWPE and SCMs with (GNPs/Fe_3_O_4_)/UHMWPE, respectively. It was shown that adding the magnetic component by decorating the GN with composites with Fe_3_O_4_ nanoparticles significantly increased the absorption contribution to shielding efficiency, due to the existence of abundant surface functional groups and lattice defects on (GNPs/Fe_3_O_4_) nanoparticles. It resulted in multiple reflections and scatterings of incident electromagnetic waves and consequently enhanced the electromagnetic absorption ability, due to the synergistic effect between Fe_3_O_4_ nanoparticles and GNPs.

## Figures and Tables

**Figure 1 materials-17-02808-f001:**
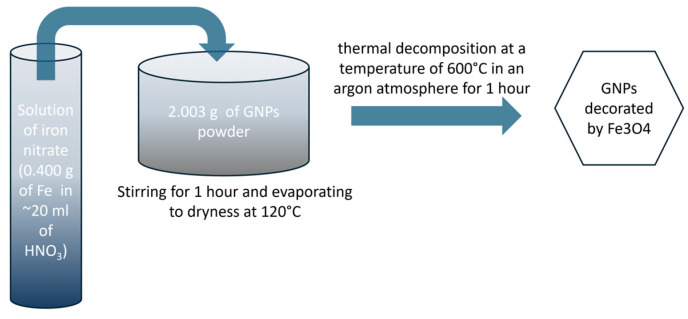
The scheme of preparation of GNPs decorated by Fe_3_O_4_.

**Figure 2 materials-17-02808-f002:**
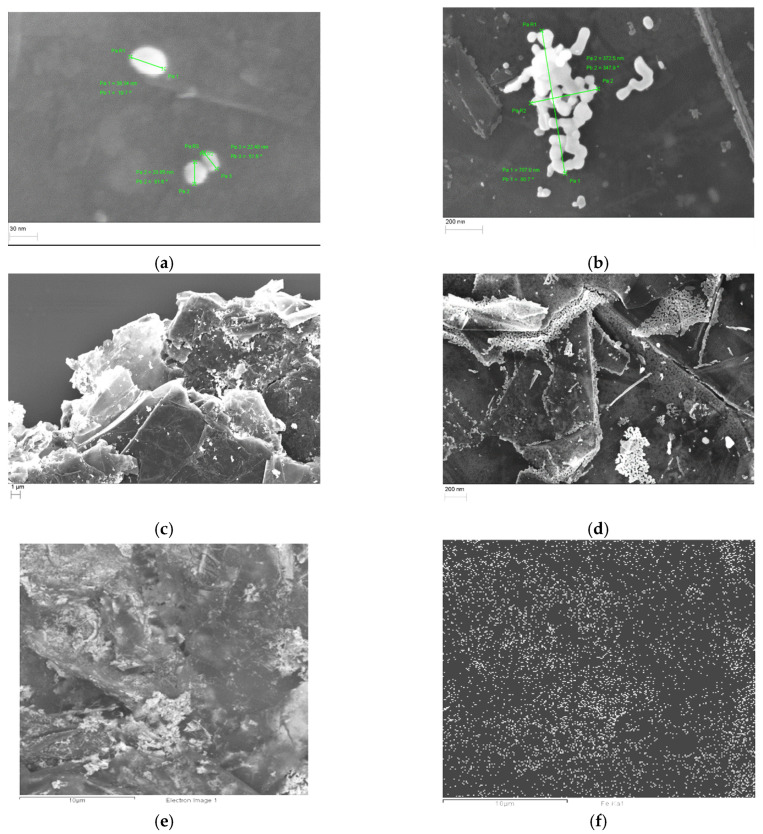
SEM images of GNPs decorated by Fe_3_O_4_ at different magnifications: (**a**) ×500,000; (**b**) ×100,000; (**c**) ×5000; and (**d**) ×60,000; (**e**,**f**) is an electron image corresponding to EDS maps for Fe.

**Figure 3 materials-17-02808-f003:**
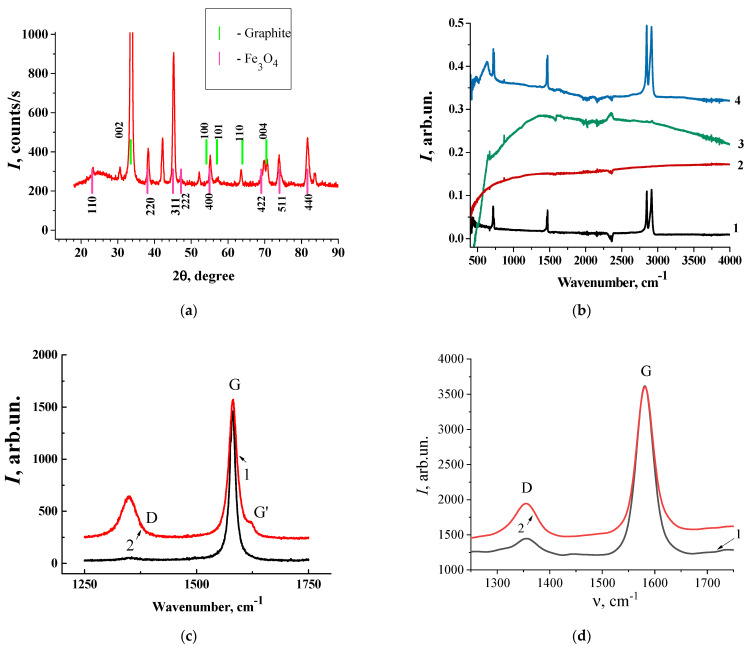
Material characterization of the samples: (**a**) XRD pattern of GNPs decorated by Fe_3_O_4_; (**b**) FTIR spectra of polyethylene (1), GNPs (2), GNPs decorated by Fe_3_O_4_ (3), and SCMs with 5 vol% (30%/Fe_3_O_4_/GNPs)/UHMPE (4); (**c**) Raman spectra of GNPs; (**d**) Raman spectra of GNPs decorated by Fe_3_O_4_ in different places of the sample.

**Figure 4 materials-17-02808-f004:**
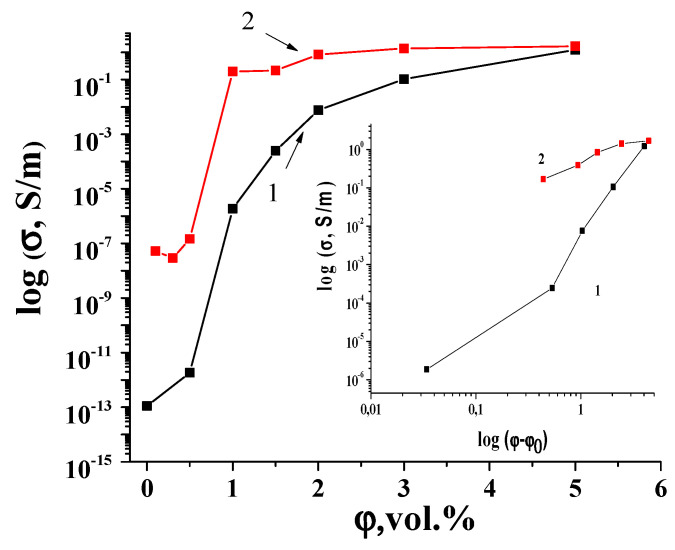
Concentration dependences of conductivity for the SCMs with GNPs/UHMPE (1) and SCMs with (GNPsFe_3_O_4_)/UHMPE (2).

**Figure 5 materials-17-02808-f005:**
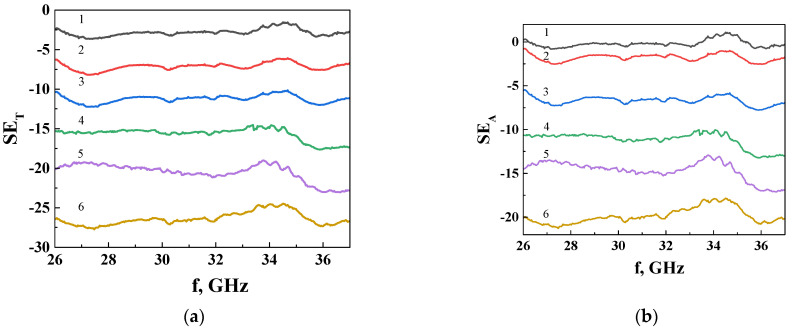
Frequency dependences of: (**a**) SE_T_ and (**b**) SE_A_ for SMCs with (GNPs/Fe_3_O_4_)/UHMWPE. Curves 1–6 in (**a**,**b**): 1—0.5 vol% ( GNPs/Fe_3_O_4_); 2—1 vol% (GNPs/Fe_3_O_4_); 3—1.5 vol% (GNPs/Fe_3_O_4_; 4—2 vol% (GNPs/Fe_3_O_4_); 5—3 vol% (GNPs/Fe_3_O_4_); and 6—5 vol% (GNPs/Fe_3_O_4_).

**Figure 6 materials-17-02808-f006:**
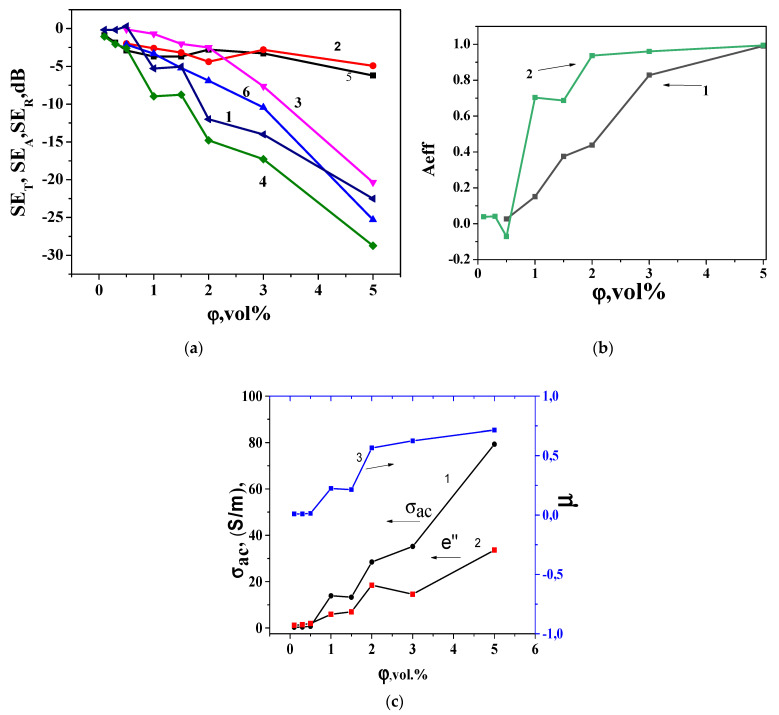
Loading dependences of (**a**) shielding efficiency SE_T_, reflection (SE_R_), and microwave absorption (SE_A_) for segregated SCMs with GNPs/UHMPE (1,2,3) and segregated SCMs with (GNPs/Fe_3_O_4_)/UHMPE (4,5,6); (**b**) shows the effective absorption index *A_eff_* for segregated SCMs with GNPs/UHMPE (1) and segregated SCMs with (GNPs/Fe_3_O_4_)/UHMPE (2); (**c**) *σ_ac_* (1), ε″ (2), and *μ_r_* (3) for composite SCMs with (GNPs/Fe_3_O_4_)/UHMPE.

**Table 1 materials-17-02808-t001:** The profile analysis of the Raman spectra of the source GNPs and GNPs decorated by Fe_3_O_4_ specimens.

Place in Sample	*x_D_*, cm^−1^	*x_G_*, cm^−1^	*I_G_/I_D_*
	GNPs		
1	1348.9	1582.0	1.64
2	1348.9	1582.0	11.11
	GNPs/Fe_3_O_4_		
1	1368	1579	3.2
2	1355.15	1581.16	2.38

**Table 2 materials-17-02808-t002:** Parameters of *φ_c_*, t, and *σ*_0_ for the studied composites.

Composite	$\varphi_{c}$ vol.%	*t*	$\sigma_{0}$, S/m
SCMs with GNPs/UHMWPE	0.97	2.78	9.78
SCMs with (GNPs/Fe_3_O_4_)/UHMWPE	0.56	1.63	4.25

## Data Availability

The raw data supporting the conclusions of this article will be made available by the authors on request.

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
