# Peer review of "Segregated Conductive Polymer Composite with Fe3O4-Decorated Graphite Nanoparticles for Microwave Shielding"

_materials, 2024, doi:10.3390/ma17122808_

Round 1

Reviewer 1 Report

Comments and Suggestions for Authors

There is a contribution to magnetodielectric losses from hybrid magnetodielectric fillers. Considering their comparative effectiveness in absorption at the targeted frequency range, why were graphene nanoplatelets (GNP) decorated with Fe3O4 instead of FeCo or NiZn ferrites? (Introduction, Paragraphs 50-56)

What methods were used to confirm the uniformity of nanoparticle decoration across batches of GNPs? Was there any variability observed?

GNPs/Fe3O4 demonstrate an improvement in EMI shielding effectiveness. What is the relationship between the frequency response of shielding effectiveness and the microstructure of the composite and its electrical conductivity? (Results, Paragraphs 287-311)

Incorporating GNPs/Fe3O4 reduced the percolation threshold significantly. How does this reduction enhance EMI shielding effectiveness and what are the underlying mechanisms that contribute to this reduction? (Results, Paragraphs 256-280):

Can you compare the cost, effectiveness, and durability of the Fe3O4-decorated GNPs composite with that of existing materials used for similar purposes?

Reviewer 2 Report

Comments and Suggestions for Authors

This reseach reported a composite of a conductive polymer with Fe3O4-decorated graphite nanoparticles for the microwave shielding application. I think that it can be accepted after revising some issues

1. All figures were not designed and organized well, such as figure labels and figure size.

2. In the introduction, the authors stated that "The use of hybrid magnetodielectric fillers contributing to magnetodielectric losses is a popular strategy for obtaining highly absorptive materials". Please adding the references to support it. Moreover, the authors need to emphasize the advatages of Fe3O4 over other magnetic materials.

3. This study used ultra-high molecular weight polyethylene. It is necessary to discuss the advantages of polyethylene as well as its applications in the literature in the microwave shielding.

4. Please provide a schematic illustration for the preparation procedure, which can be more attractive  and understandable for readers

5. The mechanism for the microwave shielding efficiency of the conductive polymer composite with Fe3O4-decorated graphite is necessary

Comments on the Quality of English Language

N/A

Round 2

Reviewer 1 Report

Comments and Suggestions for Authors

Fig.2 is not clear, and the scale is too small.

Reviewer 2 Report

Comments and Suggestions for Authors

It can be accepted for publication

Comments on the Quality of English Language

NA
